# Clinical Relevance of Uterine Manipulation on Oncologic Outcome in Robot-Assisted versus Open Surgery in the Management of Endometrial Cancer

**DOI:** 10.3390/jcm12051950

**Published:** 2023-03-01

**Authors:** Kyung Jin Eoh, Yoo-Na Kim, Eun Ji Nam, Sang Wun Kim, Young Tae Kim

**Affiliations:** 1Department of Obstetrics and Gynecology, Yongin Severance Hospital, Yonsei University College of Medicine, Yongin 16995, Republic of Korea; 2Department of Obstetrics and Gynecology, Institute of Women’s Medical Life Science, Yonsei Cancer Center, Severance Hospital, Yonsei University College of Medicine, 50-1 Yonsei-ro, Seodaemun-gu, Seoul 03722, Republic of Korea

**Keywords:** endometrial neoplasms, robotic surgical procedures, laparoscopy, mortality

## Abstract

In this study, we investigated the impact of uterine manipulation on endometrial cancer survival outcomes. We analyzed patients with endometrial cancer who underwent robot-assisted staging and open staging surgery between 2010 and 2020. Either uterine manipulators or vaginal tubes were utilized in robot-assisted staging. Propensity score matching was performed to correct baseline characteristics. Progression-free survival (PFS) and overall survival (OS) were analyzed using Kaplan–Meier curve analysis. In total, 574 patients, including those undergoing robot-assisted staging with a uterine manipulator (*n* = 213) or vaginal tube (*n* = 147) and staging laparotomy (*n* = 214), were analyzed. Propensity score matching was performed for age, histology, and stage as covariates. Before matching, Kaplan–Meier curve analysis showed that PFS and OS were significantly different among the three groups (*p* < 0.001 and *p* = 0.009, respectively). In the propensity-matched cohorts of 147 women, the previously suggested differences in PFS and OS were not observed in patients undergoing robot-assisted staging with a uterine manipulator or vaginal tube or open surgery. In conclusion, robotic surgery using a uterine manipulator or vaginal tube did not compromise survival outcomes in endometrial cancer management.

## 1. Introduction

Surgical staging is the primary step in the management of endometrial cancer. Pathological evaluation of the surgical specimen can determine the need for postoperative adjuvant treatment to minimize recurrence. In 1993, Childers et al. first reported two cases of laparoscopic-assisted vaginal hysterectomy in a patient with endometrial cancer [1]. Since then, a large volume of scientific evidence has confirmed that laparoscopic surgery offers similar clinical outcomes to those of open staging surgery but with reduced intra- and postoperative morbidity [2,3,4,5].

A variety of uterine manipulators have been developed for improved surgical performance during laparoscopic hysterectomy, including the McCartney vaginal tube [6,7,8,9]. However, several concerns arose that the use of a uterine manipulator might further disrupt the tumor and thus result in the dissemination of malignant cells. According to a recent meta-analysis, using a uterine manipulator for minimally invasive surgery (MIS) for endometrial cancer did not increase the recurrence rate [10]. Consequently, use of uterine manipulators during the process of the laparoscopic approach is considered an effective and safe procedure for patients with endometrial cancer.

Moreover, the adoption of robot-assisted laparoscopic surgery in clinical practice has increased due to its potential benefits over conventional laparoscopic surgery. Robot-assisted surgery provides improved binocular vision, seven degrees of freedom, reduced dependence on skilled assistance, and a relatively low learning curve, which has encouraged gynecologic oncologists to use MIS in the treatment of endometrial cancer [2,11,12,13,14,15,16]. However, the effectiveness and safety of utilizing a uterine manipulator in the process of robot-assisted surgery has not been thoroughly investigated in previous studies.

The objective of this study was to evaluate the oncological outcomes among patients diagnosed with endometrial cancer who underwent robot-assisted staging with a uterine manipulator, robot-assisted staging with a vaginal tube, and open staging laparotomy, after controlling for baseline characteristics through propensity score matching. The scarcity of literature evaluating the clinical significance of uterine manipulation on the survival outcome of robot-assisted management of endometrial cancer prompted the need for this study in a single tertiary care institution.

In this study, we hypothesized that staging surgery with a uterine manipulator would yield comparable oncologic outcomes to those of open staging surgery for endometrial cancer.

## 2. Materials and Methods

### 2.1. Patients

The present retrospective cohort study underwent an ethical review process and was granted approval by the Institutional Review Board (IRB) of Yonsei University College of Medicine in Republic of Korea (Ethical no. 2022-3811-001). In accordance with ethical guidelines, the IRB waived the requirement for obtaining written informed consent from study participants. The conductance of this study was guided by the principles outlined in the Declaration of Helsinki, which serve as an internationally recognized set of ethical guidelines for medical research involving human subjects.

We reviewed the medical records of patients with the Federation of Gynecology and Obstetrics (FIGO) stage I-III endometrial cancer who underwent open staging surgery or robot-assisted staging for endometrial cancer at Yonsei Cancer Center in Seoul, Republic of Korea, between January 2010 and January 2020. The medical records were analyzed to gather a range of relevant clinical variables such as demographic information (e.g., age, body mass index), histopathological information (e.g., histology), disease stage (FIGO stage), surgical details (e.g., harvested and metastatic lymph nodes, intra-operative blood loss, postoperative transfusion), and treatment information (e.g., use of adjuvant therapy).

All surgical procedures were carried out by skilled gynecologic oncology surgeons using either an open or robotic approach. The surgical staging was comprehensive and included hysterectomy, salpingo-oophorectomy, lymph node dissection (including pelvic and para-aortic nodes as required), sentinel lymph node biopsy, omentectomy, and peritoneal biopsy when necessary. To ensure the validity of the study, patients who had stage IV disease or evidence of gross extra-uterine disease at the time of surgery were excluded from the study.

Among patients receiving robot-assisted staging, either the uterine manipulator or vaginal tube was used (Figure 1a,b). Patients undergoing robot-assisted staging were positioned in the dorsal and steep Trendelenburg positions. A Foley catheter was inserted to empty the bladder. All robot-assisted staging procedures were performed using the da Vinci telerobotic system (Intuitive Surgical, Inc. Sunnyvale, CA, USA). All three systems (systems S, Si, and Xi [Intuitive Surgical Inc.]) were used during the study period. In robot-assisted staging surgery, port placement and lymphadenectomy were conducted as previously reported [11,17,18].

#### 2.1.1. RUMI Group

Procedures included an extrafascial hysterectomy with bilateral salpingo-oophorectomy, bilateral pelvic lymph node dissection, para-aortic lymph node dissection, and peritoneal cytology. The surgical assistants at the bedside and the caudal part of the patient for uterine manipulation were usually chief residents or fellows-in-training. To ensure adequate uterine handling and exposure of the pelvic space, a RUMI uterine manipulator was utilized in conjunction with a Koh colpotomy ring and a vaginal balloon pneumo-occluder (Cooper Surgical Inc., Trumbull, CT, USA).

#### 2.1.2. Tube Group

The methodology employed in the tube group was consistent with that described for the RUMI group, except for the utilization of the McCartney vaginal tube (LiNA Medical, Sydney, Australia).

### 2.2. Statistical Analysis

Our institutional follow-up strategy was designed to closely monitor patients after treatment. The follow-up protocol called for patients to be seen every 3 months for the first 2 years after treatment and then every 6 months thereafter. If disease recurrence was detected radiologically or histologically during a follow-up examination, the date of first appearance was recorded as the date of recurrence.

Progression-free survival (PFS) was defined as the interval between the date of initial diagnosis and the date of disease progression, which was determined based on the Response Evaluation Criteria in Solid Tumors (version 152 1.1) [19]. We calculated overall survival (OS) as the time between the date of initial diagnosis and that of cancer-related death or the end of the study.

All analyses were performed using R Statistical Software (v4.1.2; R Core Team 2021). Propensity score matching was used to control for baseline characteristics that could influence survival outcomes. To provide a comprehensive summary of the demographic data, descriptive statistics were calculated, including means (standard deviation) for continuous variables and frequencies (percentages) for categorical variables. Differences in patient characteristics between groups were assessed using chi-squared or Mann–Whitney U tests, and the results were reported with respect to time intervals. The PFS and OS were analyzed using Kaplan–Meier curve analysis. A significance level of *p* < 0.05 was used as the threshold for determining statistical significance in all analyses.

## 3. Results

This study included a total of 574 patients who underwent different surgical approaches for endometrial cancer staging, including robot-assisted staging with a uterine manipulator (*n* = 213), robot-assisted staging with a vaginal tube (*n* = 147), and open staging laparotomy (*n* = 214). The demographic and clinical data of these patients were analyzed, and their baseline characteristics are summarized in Table 1. There were no conversions to an open staging laparotomy among patients undergoing robot-assisted staging. Baseline characteristics, such as histology and stage, were unbalanced before propensity score matching. For instance, high-risk histology and stage III disease were more prevalent in patients undergoing staging laparotomy than in those undergoing robot-assisted surgery.

Thus, propensity score matching was performed using age, histology, and stage as covariates. After propensity score weighting was performed, 147 patients were matched in each group, and there was no difference in any of the patient characteristics except for the harvested para-aortic lymph node, estimated blood loss (EBL), and transfusion treatment. The EBL was significantly higher in the laparotomy group than in the robot-assisted surgical group (*p* < 0.001), and a significant difference was found in the proportion of patients who were treated with a transfusion between the propensity score-matched groups (*p* = 0.0002). (Table 2).

The median follow-up period after surgery was calculated to be 42.5 months (with a range of 12–71 months) in patients undergoing robot-assisted staging with a uterine manipulator, 41 months (with a range of 8–72 months) in those undergoing robot-assisted staging with a vaginal tube, and 59 months (with a range of 10–110 months) in those undergoing open staging laparotomy.

A pre-matching analysis was conducted to evaluate the rate of recurrence in the three study groups, namely the uterine manipulator group, the vaginal tube group, and the open surgery group. The results of the analysis revealed that the rate of recurrence was 8.4% in the uterine manipulator group, 4.7% in the vaginal tube group, and 15.8% in the open surgery group, with a significant difference between the groups (*p* < 0.001). The results of the Kaplan–Meier curve analysis indicated that there were significant differences in PFS and OS among the three groups (*p* < 0.001 and *p* = 0.009, respectively), as illustrated in Figure 2.

Stratified by the FIGO stage, the PFS and OS were not different between patients with stage I or II diseases, as illustrated in Figure 3.

In the cohort that underwent propensity score matching, the previously observed disparities in PFS and OS were no longer evident in the entire patient population and in each subgroup that was stratified. This result is shown in Figure 4.

## 4. Discussion

The current state-of-the-art technique in the surgical management of endometrial cancer is undergoing a transformation, with MIS increasingly replacing open surgery as the standard of care. To determine the impact of this shift, the present study was conducted to compare the survival outcomes of open staging surgery and robot-assisted staging, utilizing either a uterine manipulator or a vaginal tube. Our results revealed no significant differences in the oncological outcomes between patients who underwent robot-assisted staging with a manipulator or vaginal tube and those who underwent open surgery. To the best of our knowledge, this is the first study to evaluate the survival outcomes between open staging surgery and robot-assisted staging using either a uterine manipulator or a vaginal tube in the management of endometrial cancer.

MIS is the preferred approach for endometrial cancer treatment, as shown by results from multiple prospective randomized trials that confirm its oncological safety [7]. However, these trials did not study the use of uterine manipulators. Despite a lack of evidence that demonstrates reduced surgical complications with the use of these devices, it may be worth rethinking their use in endometrial cancer surgeries [20]. This is especially true following the unexpected results of the Laparoscopic Approach to Cervical Cancer trial, where the use of a uterine manipulator was identified as a potential contributor to negative oncological outcomes in patients with cervical cancer after minimally invasive surgery [21]. To fully understand the impact of uterine manipulators on surgical outcomes, further research through large-scale prospective clinical trials with control groups that do not use the devices is necessary.

Uterine manipulators have become a commonly used tool in various surgical procedures, particularly during hysterectomies. These instruments offer several benefits to the surgeon, including: (1) easier delineation of the fornices, making colpotomy at the end of the procedure more manageable; (2) enhanced exposure of the cul-de-sac, particularly useful in cases of adhesions or endometriosis; (3) improved ability for laparoscopic surgeons to maneuver their instruments due to the lateral mobilization of the uterus; (4) the potential to better develop the vesicouterine fold; and (5) cranial displacement of the uterus, which increases the distance between the cervix and the ureters for safer dissection and reduced risk of ureteric injury.

The rapid adoption of robot-assisted surgery has led to the growing utilization of MIS for malignancies, including endometrial cancer. This shift has been associated with a reduction in surgical complications [22,23,24,25,26]. Specifically, robot-assisted staging surgery is beneficial for patients with obesity, as it is correlated with a significantly reduced rate of surgical complications compared to that with open surgery, as well as a lower rate of conversion to open surgery when compared to conventional laparoscopy [24,27,28]. While most previous studies on the survival outcomes of MIS for endometrial cancer have compared open surgery with conventional laparoscopy rather than robot-assisted surgery, the results suggest that laparoscopy is a preferred option for patients with endometrial cancer [3,13,15]. However previous studies that focused on the issue related to the use of uterine manipulators in robot-assisted surgery are scarce.

Uterine manipulation during laparoscopic and robot-assisted hysterectomy is crucial, as the mobilization of the uterus permits a surgical approach to different tissue structures in conjunction with several types of manipulators. Various devices have been developed for appropriate uterine manipulation [6,8,20,29]. Most of these manipulators are used via insertion into the cervix and uterine cavity, while vaginal tubes are positioned extracervically and do not require cervical insertion.

Currently, there is no standard recommendation for uterine manipulation in the surgical management of endometrial cancer. Conventional manipulators inserted into the uterine cavity tend to make manipulation easier than external vaginal tubes. However, the disadvantage of uterine manipulators should be fully considered, which includes need for cervical dilatation and the difficult mobilization of large uteri. Uterine rupture is the most frequent complication, in addition to vaginal wall lacerations, excess hemorrhaging, and cervical cup melting, especially in endometrial cancer [30]. Therefore, the utilization of a vaginal tube, as an alternative to uterine manipulators, has been increasingly adopted in MIS procedures due to its advantageous characteristics that overcome the limitations posed by conventional uterine manipulators.

There are concerns that the use of a uterine manipulator might further disrupt the tumor, resulting in the spread of cancer cells. Some researchers have doubted the use of intrauterine devices during the surgical staging of endometrial malignancies. A Spanish group recently reported the adverse impact of uterine manipulation on oncologic outcomes in patients with apparent early-stage endometrial cancer [31]. By contrast, most authors have published reports suggesting that the use of uterine manipulators is not related to harmful oncologic outcomes [32,33]. Clinical and pathological studies have shown that capillary-like space invasion is an artifact [32,33]. Many large-scale studies have rejected the hypothesis that the uterine manipulator might increase the dissemination of cancer cells into the pelvic cavity during surgical staging of endometrial cancer [34,35]. An Italian gynecologic oncology group, in a retrospective multi-institutional cohort study, assessed the risk and site of disease recurrence, overall survival, and disease-specific survival in women who underwent laparoscopic surgery for endometrial cancer, with and without the use of a uterine manipulator. Propensity-matched analysis showed that the sites of recurrence were comparable between the groups. The type of manipulator and presence or absence of a balloon on the tip of the instrument were also considered in this study and showed no significant association with the risk of recurrence [36]. In addition, a recent meta-analysis suggested that the use of a uterine manipulator for MIS for endometrial cancer did not increase the recurrence rate [10]. Therefore, the use of uterine manipulator during the process of minimally invasive approach is considered an effective and safe procedure for patients with endometrial cancer.

In the present study, we compared the survival outcomes between propensity score-matched groups of patients undergoing open and robotic surgery, using either a uterine manipulator or a vaginal tube, after the introduction of robot-assisted surgery. There were no significant differences in oncological outcomes between patients undergoing robot-assisted staging with a manipulator or vaginal tube and those undergoing open staging surgery for endometrial cancer. There is a paucity of well-designed studies that have evaluated the oncologic outcomes between open staging surgery and robot-assisted staging using either a uterine manipulator or vaginal tube in the management of endometrial cancer. Only one study has addressed the impact of the use of a uterine manipulator in robot-assisted surgery for endometrial cancer regarding the prognosis. Ito et al. compared oncologic outcomes between patients who underwent robot-assisted staging with a uterine manipulator and those who underwent open staging laparotomy and found no difference in the long-term prognosis. They also suggested that the use of uterine manipulators was highly effective for safe robotic surgery because most surgeries are performed by novice surgeons owing to the rapid learning curve [37].

The current study presents several strengths that enhance its validity and reliability. Firstly, all surgical procedures and adjuvant treatments were performed at a highly respected tertiary referral institution, where patients received care from skilled and experienced gynecologic oncologists and designated radiation oncologists who have received specialized training through a fellowship program. Additionally, the study was able to include a relatively large sample size of 574 patients, which provides a robust representation of the population under investigation. Despite these strengths, there are several limitations to the study that should be considered. Firstly, the study was conducted retrospectively, meaning that the data analyzed were collected after the fact. This type of study design can limit the accuracy and completeness of the data, as well as increase the risk of confounding variables that were not measured or recorded. Secondly, there may be a potential selection bias in the study, especially given that the patients were selected based on their eligibility for robotic surgery. This can limit the generalizability of the results and may not accurately reflect the experiences and outcomes of patients who were not eligible for or did not choose to undergo this type of surgery. Given these limitations, it is important to acknowledge that further research is needed to validate the findings of the current study. This can include larger, prospective studies that are designed to control for potential confounding variables and minimize the risk of selection bias. Such studies will provide a more comprehensive understanding of the experiences and outcomes of patients undergoing robot-assisted surgical procedures using either a uterine manipulator or a vaginal tube for endometrial cancers.

## 5. Conclusions

In conclusion, our results suggest that the adoption of robotic surgery using either a uterine manipulator or a vaginal tube is a safe and effective procedure for patients with endometrial cancer. Although there is still a need for well-designed studies to further evaluate the oncological outcomes between open surgery and robot-assisted surgery using a uterine manipulator or vaginal tube, our study provides evidence to support the use of robot-assisted surgery in the management of endometrial cancer.

## Figures and Tables

**Figure 1 jcm-12-01950-f001:**
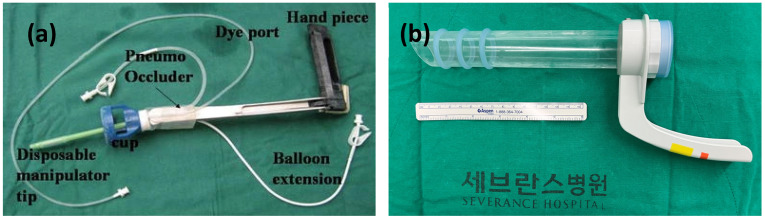
(**a**) RUMI uterine manipulator with a Koh colpotomizer system and (**b**) McCartney vaginal tube. Korean in (**b**) means Severance Hospital.

**Figure 2 jcm-12-01950-f002:**
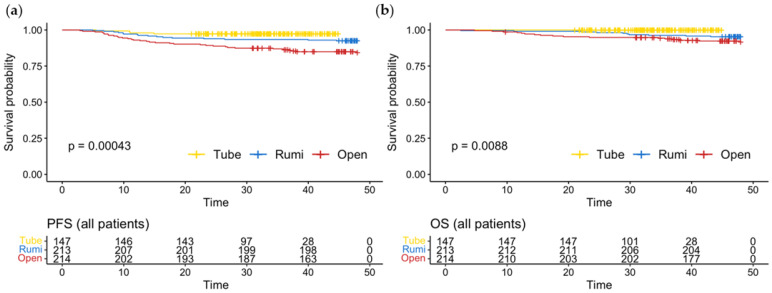
(**a**) Progression-free survival; (**b**) overall survival in the entire cohort.

**Figure 3 jcm-12-01950-f003:**
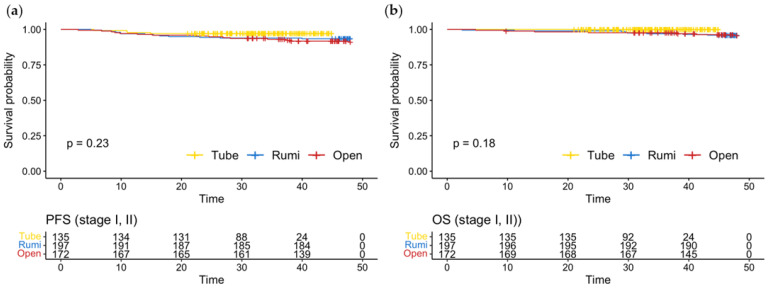
(**a**) Progression-free survival; (**b**) overall survival in patients with early stage disease (Stage I, II).

**Figure 4 jcm-12-01950-f004:**
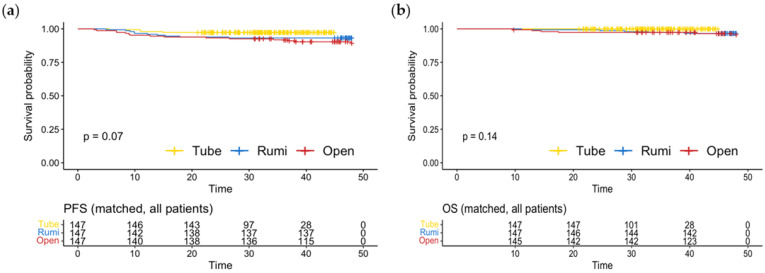
(**a**) Progression-free survival; (**b**) overall survival in the entire cohort after propensity score matching.

**Table 1 jcm-12-01950-t001:** Patient characteristics.

Characteristics	Overall	RUMI	Tube	Open	*p*
	(*n* = 574)	(*n* = 213)	(*n* = 147)	(*n* = 214)	
Age, (SD)	53.8 (9.5)	51.6 (8.6)	51.7 (9.6)	54.8 (10.2)	0.000
BMI	25.1 (5.2)	25.0 (5.4)	25.6 (5.2)	24.7 (4.8)	0.243
Histology					<0.000
Endometrioid (%)	454 (79.1)	162 (76.1)	139 (94.6)	153 (71.5)	
Carcinosarcoma	34 (5.9)	8 (3.8)	1 (0.7)	25 (11.7)	
Serous	19 (3.3)	6 (2.8)	4 (2.7)	9 (4.2)	
Squamous	21 (3.7)	20 (9.4)	1 (0.7)	0	
Clear cell	14 (2.4)	3 (1.4)	0	11 (5.1)	
Mucinous	8 (1.4)	6 (2.8)	0	2 (0.9)	
Other	24 (4.2)	8 (3.8)	2 (1.4)	14 (6.5)	
FIGO Stage (%)					<0.000
IA	398 (69.3)	168 (78.9)	109 (74.1)	121 (56.5)	
IB	78 (13.6)	23 (10.8)	21 (14.3)	34 (15.9)	
II	28 (4.9)	6 (2.8)	5 (3.4)	17 (7.9)	
III	70 (12.2)	16 (7.5)	12 (8.2)	42 (19.6)	
Pelvic LN harvested	9 (0–72)	9 (0–72)	4 (0–51)	14 (0–54)	<0.000
Para LN harvested	1 (0–51)	2 (0–36)	0 (0–9)	2.5 (0–51)	<0.000
Pelvic LN (%)					0.000
No	523 (91.1)	202 (94.9)	139 (94.6)	182 (85.0)	
Yes	51 (8.9)	11 (5.2)	8 (5.4)	32 (15.0)	
Para LN (%)					0.015
No	552 (96.2)	206 (96.7)	146 (99.3)	200 (93.5)	
Yes	22 (3.8)	7 (3.3)	1 (0.7)	14 (6.5)	
Preoperative Hb (SD)	12.3 (1.8)	12.9 (1.4)	12.1 (2.0)	11.8 (1.8)	<0.000
EBL (mL)	141.8 (313.4)	73.9 (75.3)	53.3 (57.2)	270.2 (479.3)	<0.000
Transfusion (%)					0.000
No	526 (91.6)	197 (92.5)	144 (98.0)	185 (86.4)	
Yes	48 (8.4)	16 (7.5)	3 (2.0)	29 (13.5)	
Adjuvant therapy (%)					<0.000
None	334 (58.2)	150 (70.4)	97 (66.0)	87 (40.7)	
RT	132 (23.0)	34 (16.0)	35 (23.8)	63 (29.4)	
CT	83 (14.5)	21 (9.9)	12 (8.2)	50 (23.4)	
CCRT	25 (4.4)	8 (3.8)	3 (20)	14 (6.5)	

Data are presented as the mean (SD) or *n* (%). SD, standard deviation; BMI, body mass index; FIGO, International Federation of Gynecology and Obstetrics; LN, lymph node; Hb, hemoglobin; EBL, estimated blood loss; RT, radiotherapy; CT, chemotherapy; CCRT, concurrent chemoradiotherapy.

**Table 2 jcm-12-01950-t002:** Patient characteristics after propensity score matching.

Characteristics	RUMI	Tube	Open	*p*
	(*n* = 147)	(*n* = 147)	(*n* = 147)	
Age, (SD)	51.6 (9.0)	51.7 (9.6)	53.0 (9.9)	0.404
BMI (SD)	25.0 (4.8)	25.6 (5.2)	25.1 (5.3)	0.488
Histology				0.986
Endometrioid (%)	138 (93.9)	139 (94.6)	139 (94.6)	
Carcinosarcoma	1 (0.7)	1 (0.7)	1 (0.7)	
Serous	5 (3.4)	4 (2.7)	6 (4.1)	
Squamous	1 (0.7)	1 (0.7)	0	
Clear cell	0	0	0	
Mucinous	0	0	0	
Other	2 (1.4)	2 (1.4)	1 (0.7)	
FIGO Stage (%)				0.819
IA	114 (77.6)	109 (74.1)	103 (70.1)	
IB	21 (14.3)	21 (14.3)	25 (17.0)	
II	4 (2.7)	5 (3.4)	7 (4.8)	
III	8 (5.4)	12 (8.2)	12 (8.2)	
Pelvic LN harvested	8 (0–46)	4 (0–51)	14 (0–54)	0.062
Para LN harvested	2 (0–21)	0 (0–9)	2 (0–35)	<0.000
Pelvic LN				0.743
No	140 (95.2)	139 (94.6)	137 (93.2)	
Yes	7 (4.8)	8 (5.4)	10 (6.8)	
Para LN				0.362
No	143 (97.3)	146 (99.3)	145 (98.6)	
Yes	4 (2.7)	1 (0.7)	2 (1.4)	
Preoperative Hb (SD)	13.0 (1.4)	12.1 (2.0)	11.9 (2.0)	0.071
EBL (mL)	70.9 (70.5)	53.3 (57.2)	231.8 (206.8)	<0.000
Transfusion (%)				0.000
No	138 (93.9)	144 (98.0)	126 (85.7)	
Yes	9 (6.1)	3 (2.0)	21 (14.3)	
Adjuvant therapy (%)				0.062
None	108 (73.5)	97 (66.0)	81 (55.1)	
RT	24 (16.3)	35 (23.8)	46 (31.3)	
CT	12 (8.2)	12 (8.2)	15 (10.2)	
CCRT	3 (2.0)	3 (2.0)	5 (3.4)	

Data are presented as the mean (SD) or *n* (%). SD, standard deviation; BMI, body mass index; FIGO, International Federation of Gynecology and Obstetrics; LN, lymph node; Para, para-aortic; Hb, hemoglobin; EBL, estimated blood loss; RT, radiotherapy; CT, chemotherapy; CCRT, concurrent chemoradiotherapy.

## Data Availability

Due to the nature of this retrospective study, participants of this study did not agree for their data to be shared publicly, so supporting data are not available.

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
