# Peer review of "Clinical Relevance of Uterine Manipulation on Oncologic Outcome in Robot-Assisted versus Open Surgery in the Management of Endometrial Cancer"

_jcm, 2023, doi:10.3390/jcm12051950_

Round 1

Reviewer 1 Report

This manuscript studies the impact on survival of the uterine manipulator vs vaginal tube with robotics on survival and also compares these groups with open surgery. 

The manuscript has some concerns that must be solved before publication.

Mainly the objective of the study is not clear. Could it be the impact of the uterine manipulator on survival? If so, the groups to be compared should be the robots and not the open surgery. Did the authors use open surgery as the gold standard?

In the abstract must appear the final sample after psmatch. In the conclusion sentence: “In conclusion, robotic surgery using a uterine manipulator or vaginal tube did not compromise survival outcomes in endometrial cancer management, despite considering the learning curves of all gynecologic oncologists”. The learning curve was not analyzed, so it must not be a part of the conclusion. 

Introduction: I do not agree with the sentence “Since studies evaluating the clinical relevance of uterine manipulation on the survival outcome of robot-assisted management of endometrial cancer are lacking” because there are studies and meta-analyses evaluating this. More than "lacking" should be used “scarce”.

Material and methods:  the sentence “Patients were excluded if….. or if they had already undergone a hysterectomy or lymphadenectomy for endometrial cancer” is confusing because all these patients must undergo these procedures to be included. Rewrite.

Include the median of follow-up of the sample and also the differences between the group would be recommended. 

In the results section, the cases with lymphadenectomy are few considering the cases (20%) of high-risk EC in the sample. It will be interesting to know the number of positive nodes to know how many cases IIIC FIGO stage are. In addition, the count of nodes harvested is low probably, because the mean was obtained using all samples counting the cases without lymphadenectomy. These cases must be removed when the number of nodes is shown. Did authors obtain 0 nodes in an LDN?

In figures 2, 3 and 4 the footnote is repeated. Remove one.

The text line 139 “Before matching, Kaplan–Meier curve analysis showed that PFS and OS were significantly different among the three groups (p = 0.000 and p = 0.008, respectively) (Figure 2). The p-value must be p<0.001 and p=0.009 respectively.

Line 149: However, among patients with stage III disease, PFS was different between the three groups (p = 0.07), yet OS was not different (p = 0.14). In the propensity-matched cohort, the previously suggested differences in PFS and OS were no longer observed in all patients and each stratified subgroup (Figure 4). Clarify, Does figure 4 represent the psmatching? and are the p values from figure 4 or stage III?

Author Response

This manuscript studies the impact on survival of the uterine manipulator vs vaginal tube with robotics on survival and also compares these groups with open surgery.

The manuscript has some concerns that must be solved before publication.

Mainly the objective of the study is not clear. Could it be the impact of the uterine manipulator on survival? If so, the groups to be compared should be the robots and not the open surgery. Did the authors use open surgery as the gold standard?

Response: Thank you very much for your valuable comments.

The objective of this study was to investigate the clinical impact of uterine manipulation on the survival outcome of robot-assisted management of endometrial cancer. We sought to assess oncologic outcomes of patients who underwent robot-assisted staging with uterine manipulator, robot-assisted staging with vaginal tube, and open staging laparotomy. We used open surgery as the gold standard.

Uterine manipulation during robot-assisted hysterectomy is crucial, as the mobilization of the uterus permits surgical approach to different tissue structures in conjunction with several types of manipulators. Various devices have been developed for appropriate uterine manipulation. Most of these manipulators are used by insertion into the cervix and uterine cavity, while vaginal tubes are just positioned extracervically.

Currently, there is no standard recommendation for uterine manipulation in the surgical management of endometrial cancer.

In the abstract must appear the final sample after psmatch. In the conclusion sentence: “In conclusion, robotic surgery using a uterine manipulator or vaginal tube did not compromise survival outcomes in endometrial cancer management, despite considering the learning curves of all gynecologic oncologists”. The learning curve was not analyzed, so it must not be a part of the conclusion.

Response: Propensity score-matched cohorts of 147 women in each surgical group showed no significant differences in survival. We inserted the cohort of 147 women in the Abstract. As you mentioned, the learning curve was not analyzed in this study. So, we changed the conclusion of the Abstract.

Introduction: I do not agree with the sentence “Since studies evaluating the clinical relevance of uterine manipulation on the survival outcome of robot-assisted management of endometrial cancer are lacking” because there are studies and meta-analyses evaluating this. More than "lacking" should be used “scarce”.

Response: Thank you very much for your remark. As you mentioned, we changed "lacking" into “scarce”.

Material and methods: the sentence “Patients were excluded if….. or if they had already undergone a hysterectomy or lymphadenectomy for endometrial cancer” is confusing because all these patients must undergo these procedures to be included. Rewrite.

Response: We rewrite the sentences, as follow.

All surgeries were performed using an open or robotic approach by gynecologic oncology surgeons. Patients underwent complete surgical staging, including hysterectomy, salpingo-oophorectomy, lymph node dissection (pelvic with/without paraaortic nodes), sentinel lymph node biopsy, omentectomy, and peritoneal biopsies when required. Patients with stage IV disease or gross extra-uterine disease at the time of the operation were also excluded.

Include the median of follow-up of the sample and also the differences between the group would be recommended.

Response: We include the median follow-up of the groups in the Result, as below.

The median follow-up period after surgery was 42.5 months (range, 12-71 months) in patients undergoing robot-assisted staging with a uterine manipulator, 41 months (range 8-72 months) in those undergoing robot-assisted staging with vaginal tube, 59 months (range 10-110 months) in those undergoing open staging laparotomy, respectively.

In the results section, the cases with lymphadenectomy are few considering the cases (20%) of high-risk EC in the sample. It will be interesting to know the number of positive nodes to know how many cases IIIC FIGO stage are. In addition, the count of nodes harvested is low probably, because the mean was obtained using all samples counting the cases without lymphadenectomy. These cases must be removed when the number of nodes is shown. Did authors obtain 0 nodes in an LDN?

Response: We have performed the sentinel lymph node mapping and biopsy for all patients with EC since 2017. So the cases with lymphadenectomy was few considering the high risk endometrial cancer. There were a few cases with 0 nodes after lymphadenectomy in 3 groups.

In figures 2, 3 and 4 the footnote is repeated. Remove one.

Response: We corrected the footnotes.

The text line 139 “Before matching, Kaplan–Meier curve analysis showed that PFS and OS were significantly different among the three groups (p = 0.000 and p = 0.008, respectively) (Figure 2). The p-value must be p<0.001 and p=0.009 respectively.

Response: We corrected p values.

Line 149: However, among patients with stage III disease, PFS was different between the three groups (p = 0.07), yet OS was not different (p = 0.14). In the propensity-matched cohort, the previously suggested differences in PFS and OS were no longer observed in all patients and each stratified subgroup (Figure 4). Clarify, Does figure 4 represent the psmatching? and are the p values from figure 4 or stage III?

Response: We corrected sentences. Figure 4 represents the PS matching in the entire cohort.

Reviewer 2 Report

Dear Authors, 

This article focuses on an important topic, and the results add important insights to the existing literature. 

Some recommendations:

- Table 1 – where histology was mentioned instead of 23, you will correct with 24

- Table 2 – in the group where adjuvant CCRT therapy was described, 3.4% represent 5 cases instead of one case, as mentioned.

- Enter the number and date of the ethical approval.

In lines 99-101, you defined recurrence and evaluated the recurrence rate in discussions, without presenting data regarding its rate in the results.

- What was the model and, respectively, the recurrence rate at the level of the groups?

- What was the average time to relapse?

- What was the five-year recurrence-free survival to compare with other studies?

- It would be interesting to discuss the different rates of recurrence between early versus advanced stages of endometrial cancer related to additional high-risk pathological factors.

- What was the conversion rate to laparotomy from minimally invasive procedures?

- Line 194 adding references.

Kind regards

Author Response

  1. Table 1 – where histology was mentioned instead of 23, you will correct with 24

Response: We corrected the counts. Thank you for your thoughtful comments.

  1. Table 2 – in the group where adjuvant CCRT therapy was described, 3.4% represent 5 cases instead of one case, as mentioned.

Response: We corrected the counts.

  1. Enter the number and date of the ethical approval.

Response: The ethical number is 2022-3811-001. The date: January 6th, 2023.

  1. In lines 99-101, you defined recurrence and evaluated the recurrence rate in discussions, without presenting data regarding its rate in the results.

  1. What was the model and, respectively, the recurrence rate at the level of the groups?

  1. What was the average time to relapse?

Response: Before PS matching, the rate of recurrence was 8.4% in the uterine manipulator group, 4.7% in the vaginal tube group, and 15.8% in the open surgery group (P<.001). We included these findings in the Results.

  1. What was the five-year recurrence-free survival to compare with other studies?

Response: The estimated 5-year PFS and OS rates were 87.6% and 91.0% in the entire cohorts, respectively. These findings were compatible to other studies on survival of endometrial cancer.

  1. It would be interesting to discuss the different rates of recurrence between early versus advanced stages of endometrial cancer related to additional high-risk pathological factors.

Response: In the open surgery group, a large number of stage III patients were included, and the rate of recurrence was higher than in other groups due to additional high-risk pathological factors. So, PS matching was needed to adjust this. We included these findings in the Results.

  1. What was the conversion rate to laparotomy from minimally invasive procedures?

Response: There were no conversions to an open staging laparotomy among patients undergoing robot-assisted staging.

  1. Line 194 adding references.

 Response: We added references in the line 194.

Round 2

Reviewer 2 Report

The objectives proposed by the authors in this study were achieved, and the changes were made according to the recommendations.

Kind regards